# Analytic Method for Optimizing the Allocation of Manure Nutrients Based on the Assessment of Land Carrying Capacity: A Case Study from a Typical Agricultural Region in China

**Jingjing Sun** [1], **Xinyu Mao** [1,*], **Hiba Shaghaleh** [2], **Tingting Chang** [1], **Runzhi Wang** [3], **Senmao Zhai** [1] and **Yousef Alhaj Hamoud** [4,*]

1   College of Agricultural Science and Engineering, Hohai University, Nanjing 210024, China
2   College of Environment, Hohai University, Nanjing 210098, China
3   Institute of Animal Husbandry and Poultry Science, Nanjing 211000, China
4   College of Hydrology and Water Recourses, Hohai University, Nanjing 210098, China
*   Correspondence: 20190062@hhu.edu.cn (X.M.); yousef-hamoud11@hhu.edu.cn (Y.A.H.)

**Abstract:** The separation between planting and breeding results in an unbalanced distribution of the regional livestock and poultry manure (RLM) industry, and it has raised great concerns. A holistic analysis and problem-solving scheme using 72 townships as the research point was developed in this study. On the basis of a survey from a typical agricultural region in China, the local characteristics of manure discharge, land use, and crop cultivation were analyzed. The assessment of land carrying capacity and environmental risk assessment was conducted by simulating the nitrogen cycle. Afterwards, optimized livestock breeding strategies and inter-regional transfer and flow scheme of manure nutrients were proposed. The spatial distribution of RLM in terms of pig manure equivalent showed an imbalance of high north–south and low middle, and the nitrogen requirement of crops showed a decreasing trend from north to south. In some townships, the environmental risks were higher than level I, which indicated that pollution existed around large construction sites and water areas in the northwest. The land carrying capacity index calculated at 50% nutrient ratio displayed no overloaded risk, whereas 10–20% nutrient ratio exhibited overloaded risk. Assessments showed that the residual RLM and its nitrogen volume were 151,700 and 3574.64 tons per year, respectively. More than 80% of the study area could be used as a nitrogen nutrient sink area, and only six townships are nitrogen nutrient sources. Therefore, optimizing the allocation of manure nutrients is expected to avoid agricultural contamination from livestock manure.

**Keywords:** livestock and poultry breeding; land carrying capacity; environmental risk; nitrogen load and transportation

## 1. Introduction

The separation between planting and breeding industries [1] and unreasonable regional utilization of livestock manure [2] have caused environmental issues in China. In 2017, the cumulative total emissions of different livestock and poultry manure were $1.64 \times 10^9$ t (FW) [3]. The contribution of mass load for the manure was mainly concentrated in the eastern coastal areas of China, with Henan, Shandong, and Tianjin contributing the largest mass load [3]. The excessive investment of enterprises in waste treatment equipment makes them reluctant to spend energy on the planting industry [1], thus causing an imbalance between the planting and breeding industries. Yan et al. [2] evaluated the spatial distribution of livestock and poultry farms in a certain region and found that 50.8% of livestock and poultry farms were built in unsuitable areas, which were represented by less farmland and more residential land. Without accurate configuration, the long-term excessive application of livestock manure in arable land could cause non-point source pollution, which threatens the agriculture ecology and human health [4–6]. Pollutants,

such as nitrogen and phosphorus, contained in livestock manure may cause pollution to water bodies through processes such as surface runoff and leaching. A study [6] has shown that livestock and poultry farming is the main source of COD, TN, and TP, and it has a large contribution to agricultural non-point source pollution. Allocating manure resources rationally should comprehensively consider the regional breeding density and planting distribution, thus more research should be conducted.

Environmental risk assessment and carrying capacity analysis of arable land are the most effective methods for guiding the reutilization of livestock and poultry manures. They are direct indicators for the environmental risk assessment of livestock and poultry manure discharge to measure whether the local production of livestock and poultry manure exceeds the environmental bearing range. At present, the study of environmental risks is based on the integrated assessment of models and the assessment of single objects. The models mainly consider the differences in carrying loads of cropland and livestock manure pollution loads in different regions and classify different risk levels [7]. The assessment of a single object mainly includes the environmental risk assessment of heavy metals, antibiotics, and nitrogen and phosphorus [3,6,8]. Studies [9–11] have shown that pig manure application increased the content of Cu and Zn in the active state in soil [9], and the long-term application of livestock manure promoted the coexistence of antibiotic resistance genes and metal resistance genes in plasmids and primary integrons [10,11]. Therefore, a comprehensive environmental risk assessment based on the agricultural application of livestock and poultry manure is crucial.

Livestock land carrying capacity refers to the maximum amount of livestock that an arable land could bear [12]. Many scholars have studied the combination of livestock farming and land carrying capacity [12–15]. Yang et al. [15] analyzed the land carrying capacity and water environment risk of livestock and poultry in terms of the flow of nitrogen and phosphorus nutrients in the integrated crop–livestock system. Ning Zhu and Bo Cao et al. [14] analyzed the land carrying capacity of livestock and poultry farming in China on the basis of the nutrient supply of livestock and poultry manure and the nutrient demand of crops. However, studies on livestock land carrying capacity have rarely been linked to cropland-environment risk, and their intrinsic associations are unclear. Whether environmental risks exist when the land carrying capacity is not overloaded, and if they exist, how to resourcefully allocate livestock manure are the current issues to be considered. Meanwhile, at a municipal level, only county-scale studies exist, and small-scale studies, such as village and town-scale studies, are still blank. Using small-scale data to analyze the overall variation may be beneficial to alleviate the current dissociation status between breeding and farming. In addition, it may offer a proper method to guide the utilization of livestock manure without causing environmental pollution.

Therefore, a holistic analysis and problem-solving scheme using townships as the scale was developed in this study. By comparing the characteristics of manure discharge, land use, and crop cultivation status among different regions, the land carrying capacity and environmental risk of livestock breeding were measured and evaluated mainly on the basis of nitrogen load. Then, optimized livestock breeding strategies and a possible scheme of inter-regional transfer of nitrogen nutrients from livestock manure were proposed. The results of this study were expected to provide guidance for nutrient cycling in the crop–livestock system and the sustainable development of ecological circular agriculture.

## 2. Materials and Methods

### 2.1. Study Area and Data Sources

The study area is located in Central Jiangsu province, China, between 32°01′57″– 33°10′59″ N latitude and 119°38′21″–120°32′20″ E longitude (Figure 1), with a subtropical humid monsoon climate. The average annual temperature is 15.4 °C, the average annual rainfall is 1037.3 mm, the average sunshine duration is 2056.2 h, and the annual frost-free period is 230 d [16]. The total cultivated land area of the study area is 295,300 hm$^2$, of which the paddy field is 241,300 hm$^2$, accounting for 82% of the total cultivated land. In

2018, the number of breeding farms (households) in the study area reached more than 7000, including 15,117,500 poultry and 1,343,800 live pigs [16].

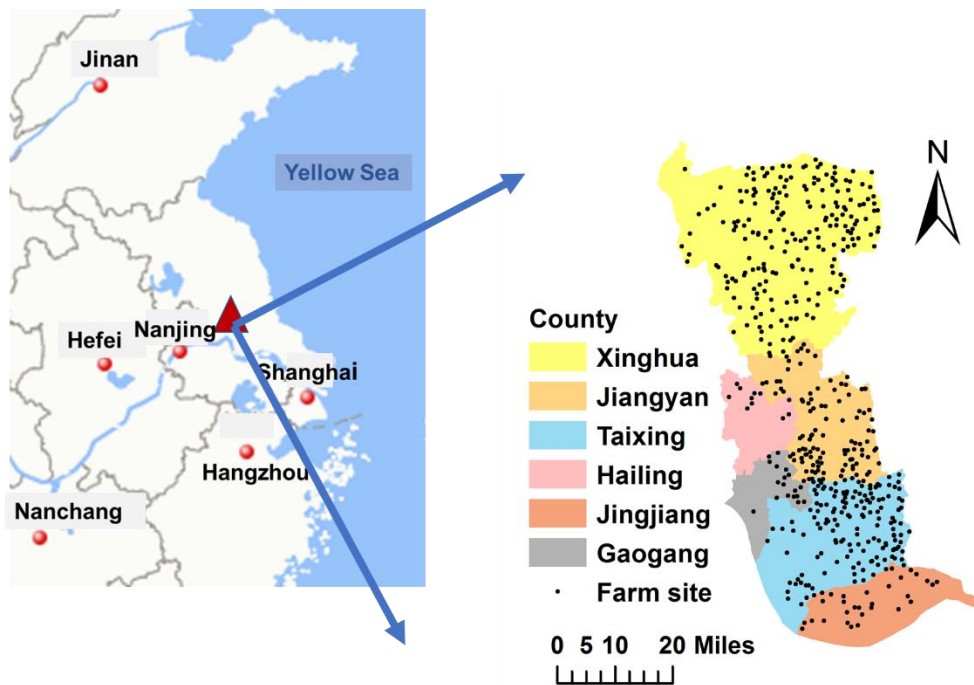

**Figure 1.** Geographical location of the study area and distribution of breeding farms.

In this study, the data included livestock and poultry breeding information (number of farms, breeding type, and quantity of livestock and poultry), cultivated area, typical crop types, and nitrogen demand. The breeding information came from the research group's preliminary investigation, and the distribution of breeding farms is shown in Figure 1. The cultivated land areas were obtained from the survey of farms supporting farmland area and the data queried in the Jiangsu government service system, statistical yearbook, and the official website of each township. Data on typical crop types and nitrogen demand were obtained from the region's statistical yearbook and China National Knowledge Internet.

### 2.2. Characteristics of Manure Discharge

The number of different kinds of livestock and poultry was converted in accordance with the number of pig equivalent to compare regional differences in the emissions of different livestock and poultry manures at the same standard. Then, the formula involved in "Environmental Problems and Countermeasures Arising from Livestock and Poultry Breeding Industry in China" [7] was used to calculate the emission of livestock and poultry manure in terms of pig manure equivalent. The conversion of pig equivalent was based on converting the fecal discharge coefficient of different kinds of livestock and poultry (1 pig = 30 laying hens; 60 broilers; 1/10 cows; 1/5 head of beef cattle; and 1/3 head of sheep). The emissions of different livestock and poultry manures were calculated using the following formula:

$$M_{ab,i} = \frac{Ei \times Ci \times D}{1000},$$

where $M_{ab,i}$ is the annual emission of livestock and poultry manure, and in this study, using the emissions of pig manure equivalent of livestock and poultry manure as the standard (including solid feces and fecal water, tons per year (t/a)); $i$ represents different types of livestock and poultry; $Ei$ is the number of livestock and poultry (head); $Ci$ is the daily excretion coefficient of different kinds of livestock and poultry (kg/d), in which the daily excretion coefficients of live pigs were used in this study, and the values of solid feces and

fecal water were 2 and 3.3 kg/d, respectively; and *D* is the number of days for livestock and poultry breeding (d), and the value is 365 d.

### 2.3. Cultivated Land Carrying Risks

Returning livestock manure to agricultural field as fertilizer is the most practical and popular pathway. The application amount should be determined in accordance with the area of farmland. According to the approach introduced in "Environmental Problems and Countermeasures of Livestock and Poultry Breeding Industry in China" [7], the loads of livestock and poultry manure in each region were estimated in township scale. The calculation formula is as follows:

$$q = \frac{Q}{S} = \sum (X \times T)/S,$$

where *q* is a load of livestock and poultry manure in terms of pig manure equivalent [t/(hm$^2$ × a)], *Q* is the total emission of pig manure equivalent of livestock and poultry manure (t/a), *S* is the effective cultivated land area (hm$^2$), *X* is the emission of different kinds of livestock and poultry manure (t/a), and *T* is the conversion factor of different kinds of livestock and poultry manure into pig manure equivalent.

The load of livestock and poultry manure could be an indirect indicator to reflect whether the density or the distribution of the breeding enterprise in a region is reasonable. Due to the different farmland areas, crop types, and land carrying capacity in each place, the environmental bearing degree of livestock manure load varies greatly [7]. Through the risk classification of the environmental bearing degree of livestock manure load (Table 1), the risk potential of farmland with the application of certain livestock manure could be evaluated. The formula for calculating the risk warning value is as follows:

$$r = q/p,$$

where *r* is the bearing degree of livestock and poultry manure pollution load in various places; *q* is a load of animal manure in pig manure equivalent [t/(hm$^2$ × a)]; and *p* is the maximum suitable application rate of organic fertilizer in local farmland (in terms of pig manure equivalent, t/(hm$^2$ × a)), and the *p* value is generally between 30 and 45 t/hm$^2$. The value was 45 t/hm$^2$ in this study.

**Table 1.** Risk value classification of manure pollution load from livestock and poultry.

| Risk Value r | ≤0.4 | 0.4–0.7 | 0.7–1.0 | 1.0–1.5 | 1.5–2.5 | >2.5 |
|---|---|---|---|---|---|---|
| Level | I | II | III | IV | V | VI |
| Environment threat | without | slight | existing | slightly serious | serious | very serious |

### 2.4. Carrying Capacity of Arable Land in Terms of Nitrogen Load

In current studies, nitrogen and phosphorus are generally used as quantitative indicators for evaluating the carrying capacity of cropland [12,14,15], and the supply and demand of nitrogen and phosphorus from livestock manure and crop are calculated to determine whether the emissions of livestock manure are overloaded. In the present paper, nitrogen was selected as quantitative indicator because the changes in nitrogen and phosphorus contents have the same pattern in different regions [17]. The calculation formulas are referred to the "Technical Guide to Land Carrying Capacity Calculation" [18]. The formula of nitrogen supplied from livestock and poultry manure is as follows:

$$Fi = M_{ab,i} \times MP_i \times PL \times PC,$$

where *Fi* is the supply of nitrogen nutrients from livestock manure (t/a); $MP_i$ is the nitrogen content of manure (g/kg); *PL* is the nutrient collection rate, and according to the "guidance" [18], the value of nitrogen is 87.5%; and *PC* is the retention rate of nitrogen, phosphorus, and other nutrients after treatment, and the value of nitrogen is 66%.

The formula of nitrogen demanded by crops from livestock and poultry manure is as follows:

$$A_{n,m} = \frac{\left(\sum P_{r,i} \times Q_i \times 10^{-2}\right) \times FP \times MP}{MR},$$

where $A_{n,m}$ is the crop demand for livestock nutrients (t/a); $P_{r,i}$ is the total production of the $i$th crop (t/a); $Q_i$ is the amount of nitrogen (phosphorus) required to produce 100 kg of the $i$th crop; $FP$ is the proportion of nutrients provided by the fertilizer out of the total nutrient requirement of the crop, with a value of 45%; $MP$ is the demand for manure nutrients as a percentage of total nutrients, generally 50%; and $MR$ is the seasonal conversion rate of manure, with values of 27.5% and 34.5% for nitrogen and phosphorus, respectively.

The livestock land carrying capacity index could be calculated as follows:

$$I = \frac{Fi}{A_{n,m}},$$

where $I$ represents the land carrying capacity index. If the $I$ value exceeds 1, the local livestock breeding is overloaded and needs to be controlled; if the $I$ value does not exceed 1, the load is not overloaded.

### 2.5. Manure Allocation Based on Nitrogen Load

In this paper, the redistribution scheme of manure nutrients was presented to realize the rational allocation of livestock and poultry manure. It refers to a strategy of excess manure nutrient flow to crop demand areas. On this basis, a formula to calculate the nitrogen (phosphorus) demand potential of arable land for livestock and poultry manure was designed to understand the sources and sinks of livestock and poultry manure.

The calculation formula for nitrogen (phosphorus) demand potential of arable land is as follows:

$$A_{N/P} = \left(S - \frac{M_{ab,i}}{r_w \times P}\right) \times S_r,$$

where $A_{N/P}$ is the amount of nitrogen (phosphorus) that the cultivated land still needs after completely absorbing the local livestock and poultry manure (t/a); $r_w$ is the risk value of non-environmental threat of manure load, with a value of 0.4; $\frac{M_{ab,i}}{r_w \times P}$ is the cultivated land area required for risk-free return of livestock and poultry manure ($hm^2$); and $S_r$ is the nitrogen (phosphorus) demand per unit cultivated land area of a region calculated in accordance with different crop species ($kg/hm^2$).

### 2.6. Analysis Methods

All data were processed using Excel 2016. All statistical analysis data were calculated using the above formulas. Statistical analysis charts were drawn using ArcGIS 10.6 and Origin 8.

## 3. Results

### 3.1. Characteristics of Breeding, Manure Emission, and Land Use in the Study Area

The emissions of livestock and poultry manure in terms of pig manure equivalent were detected to be higher in the southern and northern part of the study area. In some townships, which were located in southeast and northwest of the study area, the annual emissions of manure even reached more than 70,000 t (Figure 2a). The annual emissions of pig manure in the study area were the highest, with total solid feces and fecal water reaching 339,000 and 559,000 t/a, respectively, followed by laying hens, dairy cows, and broilers, with annual manure emissions of 592,000, 361,000, and 94,000 t, respectively (Figure 2c). The results showed that livestock farming in the region is mainly dominated by pigs, chickens, and dairy cows.

The land-use types in this area are mainly divided into three types: crop land, construction land, and water body land (Figure 2b). Among them, crop land has the largest area, accounting for more than 60% of the entire region. The area of construction land and water bodies are comparably small. The area is covered by agricultural land, mainly distributed in Taixing and Xinghua districts. The construction land is mainly distributed

in the southern part of the region, whereas the water bodies are mainly distributed in the southern border and the northern part of the region. In addition, high emissions of livestock manure were found around large construction sites and the northern watershed.

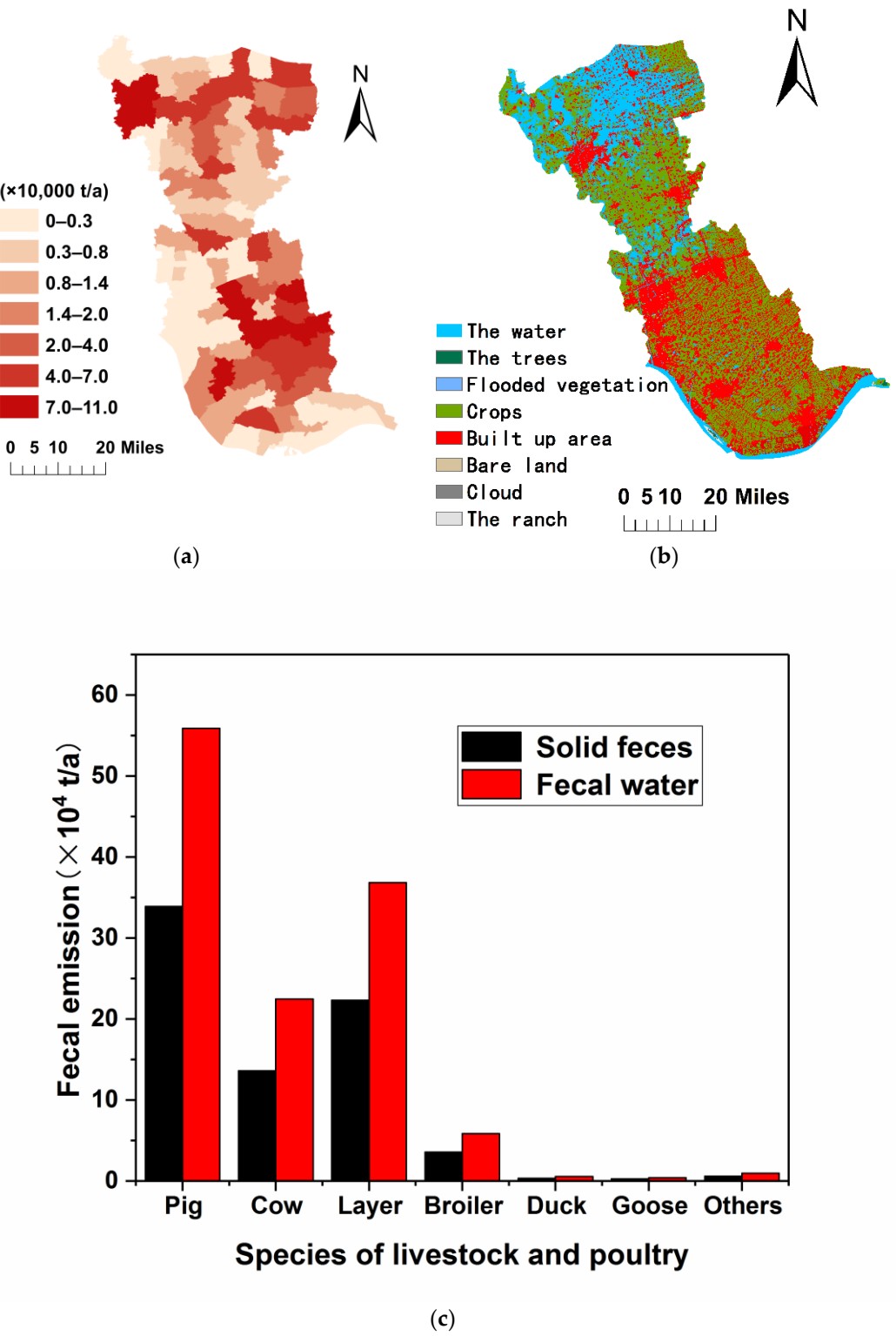

**Figure 2.** (**a**) Livestock and poultry manure emissions in township scale (in terms of pig equivalents); (**b**) land use in the study area; (**c**) manure emissions of different types of livestock and poultry.

### 3.2. Assessment of land Carrying Capacity Based on Pollution Load and Nitrogen Load
Assessment of Land Carrying Capacity Based on Manure Pollution Load

The manure pollution load in the study area was highest in the southeast region in Taixing County (Figure 3a), followed by Jiangyan County and the northern part of Xinghua County. More than 80% of the study areas had a manure pollution load of less than 20 t/hm$^2$. The manure pollution load exceeding 30 t/hm$^2$ was found in the Taixing district.

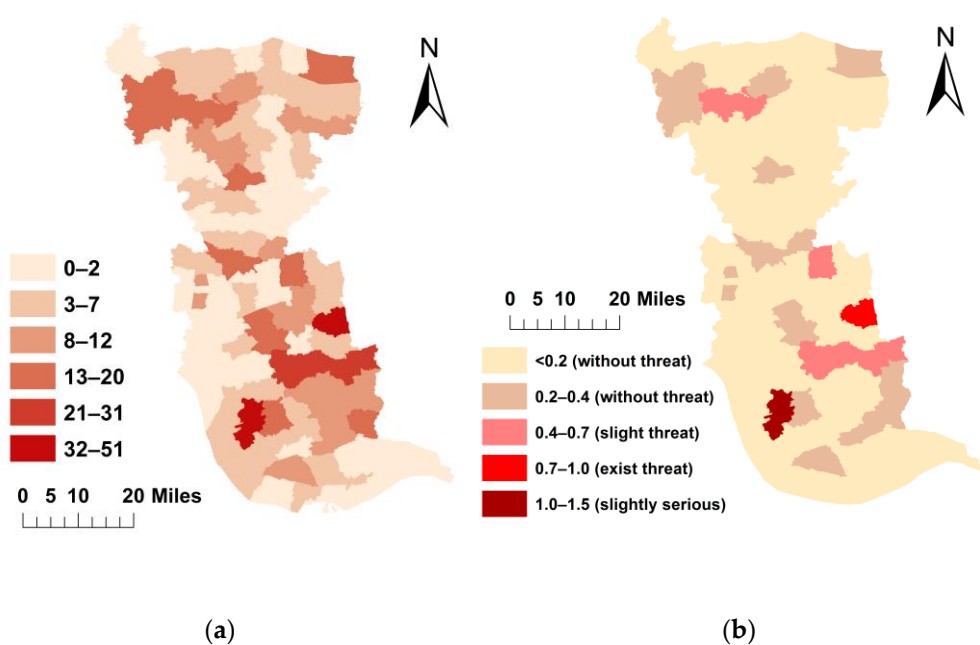

(**a**)               (**b**)

**Figure 3.** Assessment of land carrying capacity based on contaminate load at small scale: (**a**) manure pollution load (t/hm$^2$); (**b**) carrying risk level of arable land.

Under township-scale analysis, the assessed risk value R that exceeded 0.4 (level 1 risk threshold) appeared in the north and southeast areas (Figure 3b). This finding indicated an environmental threat from livestock manure discharges in these townships. The environmental threat in the northern watershed region was relatively slight but much greater in some southeast area, with risk values above 0.7 and risk levels above level 3. In addition, most of these risk areas were distributed around large construction sites.

### 3.3. Assessment of Land Carrying Capacity Based on Nitrogen Load

Crop cultivation type and yield could directly affect the magnitude of the land carrying capacity index. The main local crop types were rice, wheat, and vegetables (Table 2). Among them, vegetables had the largest production, followed by rice, wheat, melons, and oilseeds. The Taixing district was the region with the largest vegetable production, which is over 990,000 t per year, followed by Jiangyan and Xinghua districts, with annual vegetable production of 900,061 and 864,635 t, respectively. The Xinghua district was the area with the largest production of rice and wheat, with annual productions of 788,389 and 458,707 t, respectively. Maize showed the lowest yield among the crops.

Based on a crop nitrogen demand data at the county scales, the amount of nitrogen required per hectare was found to be 276.7 kg. Based on this value, the nitrogen nutrient requirement of each township was further calculated. The nitrogen nutrient demand of local crops showed a decreasing trend from north to south of the study area (Figure 4b). The Xinghua district in the north had a larger crop nitrogen nutrient demand (most 1000 t/a) and crop yield than the south areas. The largest nitrogen nutrient supply towns were in Northwestern Xinghua and Eastern Taixing, where the nitrogen supply was only 300 t/a (Figure 4a), which demonstrated that the amount of nitrogen supplied by manure is much smaller than that required by crops. If the ratio of manure nutrients

to other fertilizer nutrients = 1:1, the land carrying capacity of local townships does not exceed 1 (Figure 4c), i.e., local livestock is completely within the carrying capacity of each township and is not overloaded.

**Table 2.** Types of crops grown and annual production in the study area.

| Crop Type | Jingjiang | Taixing | Xinghua | Hailing | Gaogang | Jiangyan |
|---|---|---|---|---|---|---|
| | Annual Crop Yield/t | | | | | |
| Wheat | 107,045 | 244,246 | 458,707 | 21,710 | 39,655 | 185,190 |
| Rice | 176,290 | 372,000 | 788,389 | 41,966 | 46,054 | 286,339 |
| Vegetable | 234,499 | 992,472 | 864,635 | 135,645 | 166,463 | 900,061 |
| Corn | 603 | 4641 | 8401 | 762 | 3848 | 14,252 |
| Bean | 3312 | 8210 | 9234 | 2776 | 4528 | 8528 |
| Potato | 1305 | 1405 | 2633 | 143 | 200 | 1577 |
| Oilseed | 3465 | 33,019 | 24,307 | 2308 | 4628 | 18,794 |
| Melon | 18,332 | 9229 | 87,662 | 4573 | 9978 | 17,512 |

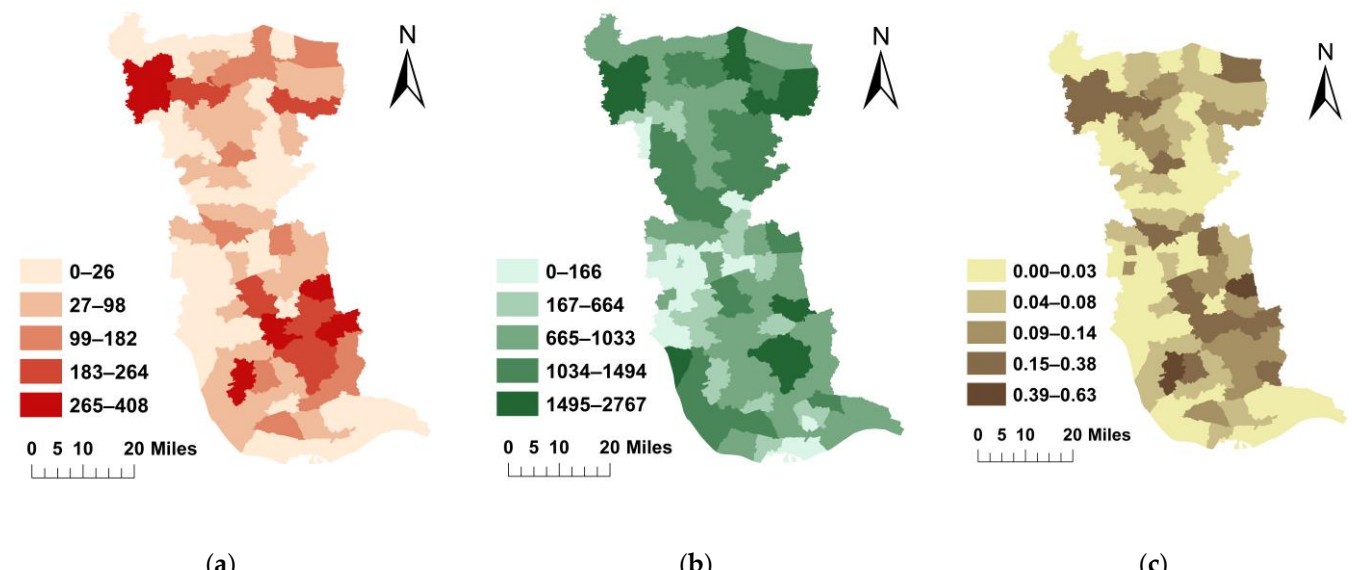

(**a**)  (**b**)  (**c**)

**Figure 4.** Township-scale (**a**) manure nitrogen nutrient supply (t/a); (**b**) crop nitrogen nutrient demand (t/a); (**c**) livestock land carrying capacity (calculated as 50% of total nutrients from livestock manure to agricultural fertilizer).

The study showed that the livestock breeding in several risk towns (Figure 3b) was not overloaded. The land carrying capacity value calculated at 50% nutrient ratio may not be a good response to the current local livestock farming situation. Thus, a smaller ratio is adopted to achieve sustainable development of the local ecology.

The calculation results showed that the land carrying capacity at 30% nutrient ratio is the valve for all townships to accommodate the current emissions of livestock manure (Figure 5a). Meanwhile, 20% and 10% nutrient ratios showed an environmental overload (Figure 5b,c). By contrast, the results calculated at 10% nutrient ratio were closest to the results of the risk evaluation on the basis of pollution load (Figure 3b).

*3.4. Livestock Multiplication Potential and Nutrient Transfer Program*
Livestock Multiplication Potential under Different Nutrient Ratios

An enormous potential for livestock multiplication occurred in the region. The results calculated at a nutrient ratio of 20–30% showed that livestock multiplication in some townships could reach 50,000 heads, with the Xinghua district being the largest (Figure 6a,b). At a nutrient ratio of 10%, the livestock multiplication in most townships in the Northern

Xinghua district could exceed 20,000 heads (Figure 6c). Most of the southern region showed an increase of more than 10,000 animals. Meanwhile, a negative growth rate appeared in several townships in the north and south.

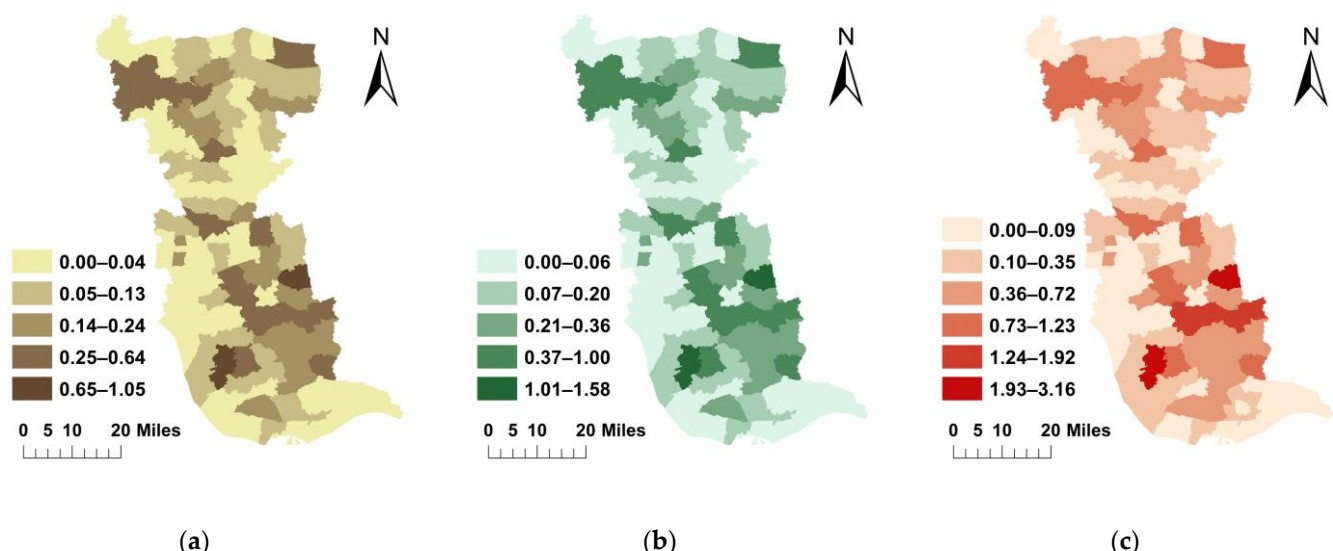

**Figure 5.** Land carrying capacity calculated with different ratios of livestock manure nutrients to total nutrients of agricultural fertilizers: (**a**) 30%, (**b**) 20%, and (**c**) 10%.

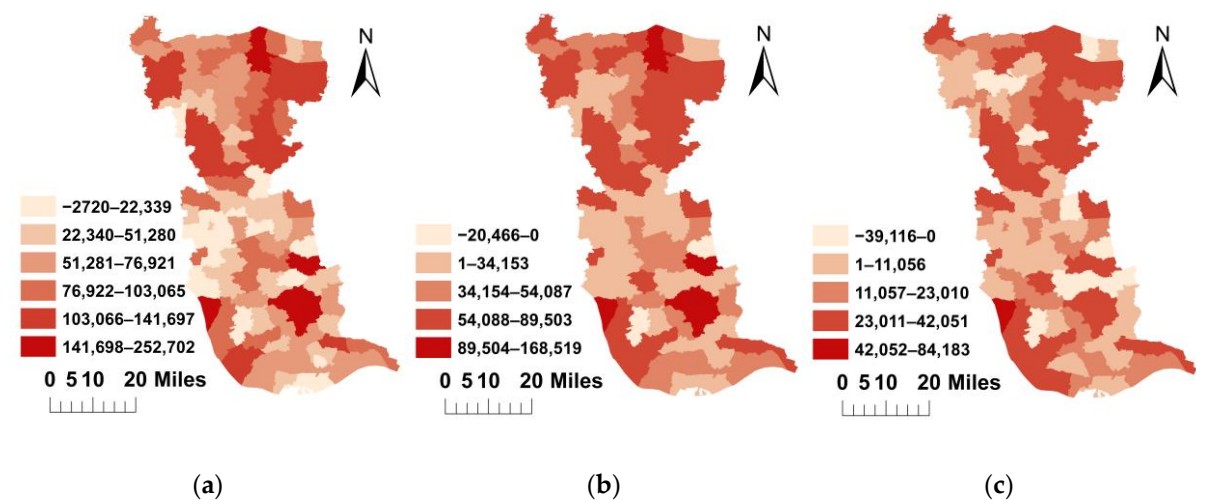

**Figure 6.** Livestock multiplication potential (head) calculated with different ratios of livestock manure nutrients to total nutrients of agricultural fertilizers: (**a**) 30%, (**b**) 20%, and (**c**) 10%.

### 3.5. Nutrient Transfer Scheme Based on Pollution Load

The assessment results of land carrying capacity (Figures 3b and 5c) demonstrated that the residual manure volume in the risk areas was about 35% of the total, which was 151,700 t/a (Figure 7). Among them, pig manure, dairy cattle manure, and chicken manure (including egg and broiler chickens) had residuals of 65,600, 41,100, and 38,900 t/a, respectively. The total surplus of nitrogen (the amount of nitrogen nutrients that could not be consumed by arable land) was 3574.64 t/a, including 1464.62 t/a for swine and 800–1000 t/a for dairy cattle and chickens.

For the sustainable development of local livestock farming, this manure should better be rationally distributed within the region. As shown in Figure 8, most of the areas (more than 80%) had positive values of nitrogen demand in cultivated land, mostly in the northern and southwestern edges. The areas with excessive nitrogen supply were mainly distributed

in two northern and four southern locations (shown at the end of the arrow in Figure 8). A nutrient redistribution could be carried out in accordance with the arrow guidance.

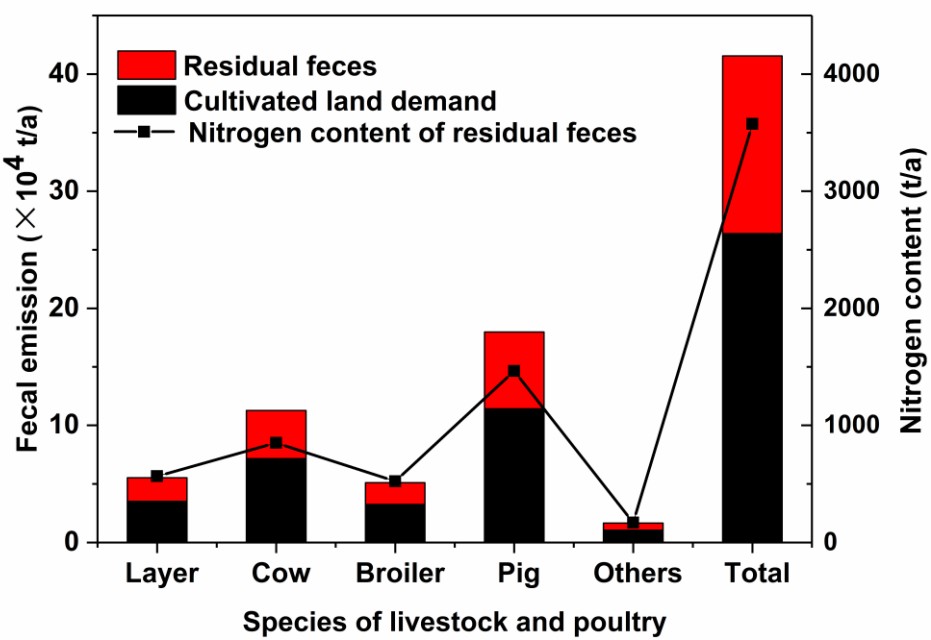

**Figure 7.** Livestock manure and nitrogen nutrient residuals.

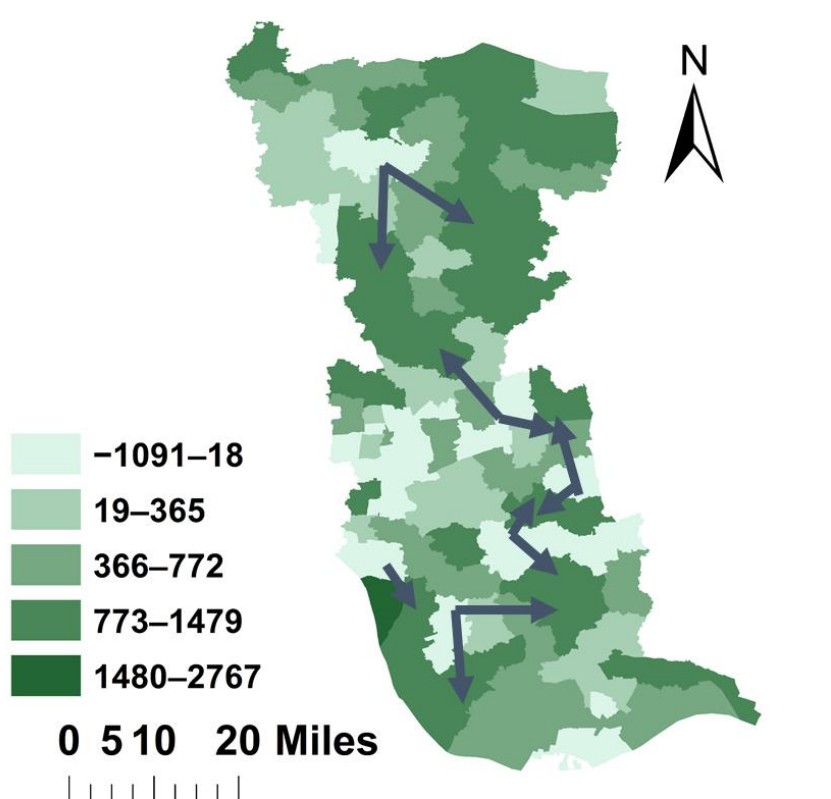

**Figure 8.** Nitrogen demand potential (t/a) for arable land based on pollution load (arrows point to livestock nutrient transfer scenarios for the township).

*3.6. Nitrogen Nutrient Flow from Animals to Farmland*

Two pathways of nitrogen nutrient flow were demonstrated (Figure 9). A portion of the nitrogen nutrients flowing from animals into manure enters the livestock farm manure

treatment system to form organic fertilizer that goes to farmland. The remainder goes to the livestock manure treatment plant for pretreatment via a transport truck and pipeline system. The nitrogen nutrients from manure then flow into the fertilizer distribution station, where they are involved in the processing, synthesis, and distribution of organic fertilizer. Later, the desired sales system accepts organic fertilizer and then flows to the cropland. So far, the distribution of nitrogen nutrients in livestock manure could be achieved.

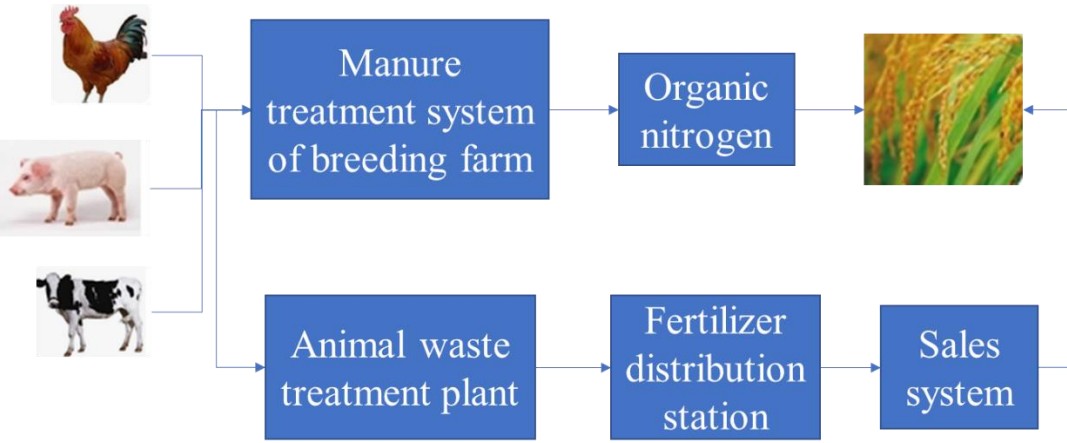

**Figure 9.** Nitrogen nutrient flow from animals to farmland.

## 4. Discussion

### 4.1. Characteristics of Breeding, Manure Emission, and Plantation

With the continuous development of the livestock and poultry farming industry, the scale of livestock and poultry farms gradually tends to be large scale and intensive [7]. The rapid intensive development of livestock and poultry farming contributes to a large amount of livestock and poultry waste discharge. In general, the spatial distribution of livestock and poultry manure in terms of pig manure equivalent in the study area is uneven, showing a high level in the north and south and a low level in the middle area. The result is consistent with the distribution patterns of livestock and poultry farms in Figure 1, as explained by the fact that the city center of Taizhou is located in the middle area (Hailing and Gaogang). According to the characteristics of livestock and poultry farming nowadays, livestock and poultry farms tend to be distributed in the suburbs and new urban areas of the city [7], which are often close to the main urban areas and have certain farmland, which helps reduce transportation costs and maximize economic benefits.

In addition, high emissions of livestock manure in terms of pig manure equivalent were found around large construction sites and the northern watershed, causing environmental hazard. This phenomenon could also be explained by the distribution characteristics of livestock farms in China, where 70% of the livestock farms are distributed in the eastern and suburban areas [19,20]. Livestock farms tend to be located in suburban and new urban areas to deliver enough meat, eggs, and milk to urban centers while reducing transportation costs [7]. However, suburban areas tend to have small farmland areas. The manure produced by livestock is sometimes not effectively disposed of and is abandoned, leading to various environmental problems [2,19,20].

Overall, the intensification of the livestock and poultry industry in the study area is dominant, which plays a great role in supporting the economic development of Jiangsu province. The study area is located in the center of Jiangsu province [21], which possesses a great agricultural foundation and hence acquired superior support from the government in livestock breeding development.

### 4.2. Assessment of Land Carrying Capacity Based on Pollution Load and Nitrogen Load

Considering the main utilization approach of livestock manure is fertilized and returned to the field [22,23], a close relationship exists between farmland area and the pollution risk of manure. In the northern area (Xinghua district), the farmland area in some townships exceeds 8500 hm$^2$, which could explain the low pollution risk of cultivated land in these areas even though the manure emission is comparatively higher. However, a large area of water body could be observed in the northwestern part, which may hold a slight environmental risk. The reason for the small area of cultivated land in the south was due to the high urbanization that restricts the development of agriculture.

The results indicated that the availability of manure nutrients at the township scale is still in a small range. One study showed that swine manure accounted for about 0.55% of the total manure nitrogen, and phosphorus and potassium accounted for only 0.24% [24]. The livestock manure is largely made up of water, with 73% of the water content in pig manure and 85% and 75% in cattle and poultry manure, respectively [3,25]. Considering that most components of livestock manure are in the form of organic matter, additives such as crop straw are usually added to adjust the carbon nitrogen ratio and improve the compost process. The organic matter content in agricultural soils could decline due to long-term tillage. By 2013, the average content of soil organic matter in China had been reduced to 1.0%, significantly lower than the 2.5–4% level in Europe and the United States [26]. The application of breeding manure could gradually improve soil fertility and quality, but given its slow nutrient release performance, organic–inorganic blending may be an effective measure for soil improvement.

Although the land carrying capacity under 50% nutrient ratio was not overloaded in all, the risk level in some areas under the township scale exceeded level 1. By contrast, a more suitable nutrient return ratio for the agricultural application of local livestock manure was found. The combination of the two methods may determine the quantification indicator for agricultural application of livestock manure in accordance with the local situation. Therefore, the nutrient return ratio of livestock and poultry should be controlled in the range of 10–30%. A nutrient ratio of 10–20% is the best to return to the field if the pollution risk of livestock and poultry manure is considered, and a nutrient ratio of 20–30% is the best to return to the field if the current development trend of local livestock and poultry farming is taken into account.

### 4.3. Livestock Multiplication Potential and Nutrient Transfer Program

The agricultural application of livestock and poultry manure is an economical and practical method to improve soil fertility [27]. Li et al. showed that compared with those of inorganic fertilizer treatment in agricultural soil, the microbial biomass C and N contents of livestock manure treatment increased by 89% and 74%, respectively, and the soil basal respiration rate and soil microbial quotient increased by 49% and 45%, respectively [28]. However, excessive agricultural use of manure could also cause the contamination of agricultural soils [29–32]. Some studies [3,10,11,33] have shown that the main source of Cu and Zn in agricultural soils is livestock manure, and the long-term or excessive application of livestock manure could cause heavy-metal pollution [3], the production of antibiotic-resistant bacteria [33], and the migration of resistant genes [10,11] in agricultural soils. Meanwhile, under-application of organic manure may lead to increased use of inorganic fertilizers. The Chinese Ministry of Agriculture issued a "zero growth" plan for inorganic fertilizers in 2020 [34]. Some studies have shown that excessive use of inorganic fertilizers could cause soil quality degradation [35], soil acidification [36], and slumping [37]. Therefore, promoting rational application of livestock manure is necessary.

Studies have shown that most of the townships with potential environmental risks are located around large building sites because of the ease of providing adequate meat, eggs, and milk to the population. Continuously increasing the livestock around the large area of construction land is not recommended. Given its environmental risk, a 10% nutrient ratio scenario for livestock augmentation is the most prudent. Although the breeding manures

could also be utilized in a nearby cropland with a certain nutrient ratio, the manure application amount should be under the land carry capacity to avoid environmental risks. So, controlling the development of livestock farming around large construction sites is the key to reducing the local livestock carrying risk. However, the large-scale relocation or closure of livestock farms may be uneconomic, which may increase employment pressure. Considering the perspective of people's livelihood, conducting manure redistribution and strengthening farm regulation (e.g., odor control) may be more effective measures to address the current problem.

Depicting a holistic scenario from animal farming [38] could help identify strategies to redistribute manure nutrients within the region and take into account the legacy properties of limited past livestock and manure use management [17,39]. The results showed that the remaining livestock manure in the risk areas could be fully consumed by surrounding regions without the risk of environmental pollution. The remaining manure was mainly pig manure, dairy cattle manure, and chicken manure, and the transport distance was within 25 km. Some studies [40,41] have shown that the transport distance limit of liquid manure usually should be less than 10 km [40,41]. Solid manures, such as chicken manure, could be transported in comparably longer distances [42], suggesting that the separation of solid and liquid fractions from the breeding farm is a key step to make it more transportable [17].

The rational allocation of livestock manure could also be influenced by climate; humanities; spatial and temporal variation of crop cultivation; pollutants, such as heavy metals and antibiotics in manure; and spatial and temporal variation of manure discharge. Future research should explore allocation options that take into account these multiple influencing factors simultaneously. Creating a network model for simulating the spatiotemporal allocation of livestock manure is also suggested.

## 5. Conclusions

Livestock manure emissions were higher in the southeastern and northwestern parts of the study area than in the middle. However, the nitrogen requirement of crops showed a decreasing trend from north to south, suggesting that the risk of contamination may exist in the south because of the livestock and poultry waste discharge.

The risk level in some areas under the township scale exceeded level 1, mostly in the southeastern part of the study area and around the large area of construction land and water body in the northwest. Thus, continuously increasing livestock and poultry breeding in the large area of construction land and the northern water body area is not recommended.

In comparison to the result of the risk assessment, the land carrying capacity under the 50% nutrient ratio was not overloaded in all, whereas the 10–30% nutrient ratio showed overload, which considered that returning the land with a 10–30% nutrient ratio of livestock and poultry manure is best. Additionally, a livestock enrichment strategy with a 10% nutrient ratio for livestock enrichment is also recommended.

The evaluation results showed that more than 80% of the study area could be used as a nitrogen nutrient sink area, and only six townships are nitrogen nutrient sources. A nutrient transfer is needed in some areas, and the amount of nitrogen nutrients transferred is around 3500 t per year. In accordance with the development trend of the local farming industry, conducting manure redistribution and strengthening farm regulation may be more effective measures. In addition, on the basis of a combination of multiple factors, future research could consider a network model to achieve the rational distribution of manure nutrients.

**Author Contributions:** J.S.: Data collection, statistical analyses, and writing—original draft. X.M.: Writing—review and editing, project administration, funding acquisition, and supervision. Y.A.H.: Writing—review and editing. H.S.: Writing—review and editing. T.C.: Supervision and writing— review and editing. R.W.: Supervision and writing—review and editing. S.Z.: Investigation, data collection, and statistical analyses. All authors have read and agreed to the published version of the manuscript.

**Funding:** This work was financially supported by the "Youth Program of National Natural Science Foundation of China" (Grant Agreement No. 51809076) and the "Taizhou Science and Technology Support Project" (Grant Agreement No. SNY20208551, SNY20208534).

**Institutional Review Board Statement:** Not applicable.

**Informed Consent Statement:** Informed consent was obtained from all subjects involved in this study.

**Data Availability Statement:** All breeding and planting data generated or analyzed during this study are available and included in this article.

**Acknowledgments:** The authors are grateful to the sponsors for funding this study and the relevant researchers at the Taizhou Research Institute for their help with this study.

**Conflicts of Interest:** The authors declare that they have no known competing financial interest or personal relationships that could have appeared to influence the work reported in this paper.

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
