# Peer review of "Analytic Method for Optimizing the Allocation of Manure Nutrients Based on the Assessment of Land Carrying Capacity: A Case Study from a Typical Agricultural Region in China"

_agronomy, doi:10.3390/agronomy13041064_

Round 1

Reviewer 1 Report

The manuscript entitled "An analytic method for optimizing the allocation of manure nutrients based on the assessment of land carrying capacity-A case study from a typical agricultural region in China" contributes with knowledge in local areas of China. Below are some comments for consideration by the authors:

Abstract

Line 11 : " The separation between ... results on an unbalanced.."

line 17 and throughout the manuscript: what emissions are the authors referring to? Please define

line 18 - please define the unit used: t/a

Introduction

Please revise this section to clarify terms and to make it easier to understand. Some sentences need to be revised and concise.

line 30 - please be careful with some statements and terms used to refer to he use of nutrient sources and other items. 

line 38 - instead of chemical, please use inorganic

lines 38-39 - where? in the soil? Please rephrase this sentence to clarify

line 43-44- The authors refer to the use of "heavy metals and antibiotics". Please verify the terms expressing the use of those strategies in general livestock systems and whether that is still current to systems in China.

line 53 - please revise the term " people's fertilization habits". This should refer to the recommendations being provided based on research.

line 58 and throughout the manuscript- please correct the term "researches". More research or more studies should be conducted.

Line 61- please remove the ";" and start a new sentence after it.

line 86- "was" not "were"

Material and methods

Did the authors developed these parameters, or are they using parameters that are well established? If the latter, please use references to previous studies using those parameters. If they are new parameters, please use references showing how those were thought about to be developed and to back up the line of thought.

line 117 - why are the authors basing this parameter in "pig equivalent". Also, please make sure to determine what emissions the authors are referring to.

line 171 - please include a reference

lines 198-200 - Based on this statement, the authors did not conduct any statistical analysis of the response values obtained. Is that correct? 

All figures need to be revised to provide better visual aid and the legend of some need to be revised as well. Major revisions are required throughout the manuscript to improve the results and the discussion sections.

Reviewer 2 Report

The manuscript entitled „An analytic method for optimizing the allocation of manure nutrients based on the assessment of land carrying capacity-A case study from a typical agricultural region in China” by Sun Jingjing et al. describes a research which aimed to optimize livestock breeding strategies and inter-regional transfer and flow scheme of manure nutrients. I find the manuscript correctly set-up, and the topic approached could be of interest to the readers of Agronomy journal. Despite these recognition, in my opinion the entire manuscript should be reviewed, because in this present form is very poorly written and without scientific soundness.

Some comments are provided below, for the authors to consider.

The abstract should be revised for a better clarity and scientific soundness. Many phrases are incorrectly set-up; the scope of the research and the specific objectives and/or hypothesis should be more clearly formulated (both in the Abstract and at the end of Introduction).

The same comment for all the sections found in the manuscript.

Introduction: During your revision, please try to avoid making statements without specifying clearly the source of the results. For example in lines 38-41, the authors didn’t specified who delivered the results described. Further, in lines 43-45 the phrase is incorrectly set-up and should be revised. I am not sure if the phrase found in lines 71-74covers your own findings or citing literature; anyway, the authors should keep in mind that they should not include their own findings in Introduction (except for a clear presentation of the scope and objectives).

The results shouldn’t include literature overview or your personal explanation of the results, but rather a description of these results. Such comments should be introduced in Discussion section, where the authors should focus on highlighting the correlation existing between their own-work and previous findings already published.

The Conclusion should be more synthetic and provide the major findings of the research; in this form the conclusion are a merely repetition of the results.

Reviewer 3 Report

The work by Jingjing et al. entitled: An analytic method for optimizing the allocation of manure nutrients based on the assessment of land carrying capacity -A case study from a typical agricultural region in China, presents an interesting results and can be usefull for improvement of sustainable crop-livestock systems.

I have only some minor suggestions for the authors which should be regarded before publication:

  1. Section 3 – results is too long, and should be shortened. In example lines 203 – 204 fits to discussion section rather than to results, and should be relocated or removed; Similarly lines 289-294; 302-306; 311-313; 329-332;
  1. In table 2 there is no unit attached in the description section, that reader could be sure what the numbers means,
  2. In lines 152 or 154 the dot is used as a multiplication sign, while in other parts of the work a different notation is used. I think it should be standardized throughout the whole manuscript,
  3. In the introduction, the authors also write about the problem of antibiotic deposition, but I do not see any results regarding this problem in the results or discussion sections.

Round 2

Reviewer 1 Report

The revised version of the manuscript entitled "An analytic method for optimizing the allocation of manure nutrients based on the assessment of land carrying capacity-A case study from a typical agricultural region in China" needs to be adjusted on based on previous comments and revisions suggested. Although the authors modified the manuscript for some adjustments, multiple comments were not addressed. Please submit a document with "Response to reviewers" on the following submission to show and guide reviewers through the changes. Please refer to the previous list of comments suggested under the first round of reviews. 

Author Response

General response: We sincerely thank the editor and all reviewers for their valuable feedback that we have used to improve the quality of our manuscript. The reviewer comments are laid out below and specific concerns have been numbered. Our response is given in normal font and changes/additions to the manuscript are given in red text.

Response to Reviewer

Abstract

Comment 1. Line 11: " The separation between ... results on an unbalanced."

Response: Thank you for your sincere suggestions. We have added this phrase to the abstract. You can see it on line 11.

Comment 2. Line 17 and throughout the manuscript: what emissions are the authors referring to? Please define

Response: Emissions refer to the emission of livestock and poultry manure in terms of pig manure equivalent. We have defined it in the abstract, the materials and methods, the results and the discussion. You can see it on lines 18, 118, 124-126, 209, 329, and 337.

Comment 3. Line 18 - please define the unit used: t/a

Response: Thank you for your considerations. t/a refers to tons per year, and we have added the definition on lines 24 and 126.

Introduction

Comment 4. Please revise this section to clarify terms and to make it easier to understand. Some sentences need to be revised and concise.

Response: Thanks for your sincere suggestions. These suggestions are very important for the revision of this paper and the guidance of follow-up research. In this section we have added the explanations and definitions of “The separation between planting and breeding industries”, “cause non-point source pollution”, “Environmental risk assessment”, and “Livestock land carrying capacity”. For example, the definition of the concept of livestock land carrying capacity has been added as the “Livestock land carrying capacity refers to the maximum amount of livestock that an arable land could bear”. You can see it on lines 33-42, 46-48, 53-55, and 66-67. Phrases and sentences have also been revised. We checked the grammar and improved the language of the article.

Comment 5. Line 30 - please be careful with some statements and terms used to refer to the use of nutrient sources and other items.

Response: In order to clarify these statements and terms more clearly, we have added the explanations of “The separation between planting and breeding industries”, “cause non-point source pollution”. We refer to some previous research data and results to explain them more specifically. You can see it on lines 33-42, 46-48.

Comment 6. Line 38 - instead of chemical, please use inorganic

Response: Thank you for your thoughtful comments. We have changed the chemical fertilizer mentioned in the introduction to inorganic fertilizer. Because of the previous additions, the paragraph referred to this problem was too much for the introduction, so we put it in the discussion section. You can see our changes on lines 386, 394-396.

Comment 7. Lines 38-39 - where? in the soil? Please rephrase this sentence to clarify

Response: In this case, in agricultural soils. We've already added it according to your suggestions. You can see our changes on line 386.

Comment 8. Line 43-44- The authors refer to the use of "heavy metals and antibiotics". Please verify the terms expressing the use of those strategies in general livestock systems and whether that is still current to systems in China.

Response: Thank you for your constructive questions. We refer here to heavy metals and antibiotics to illustrate the ecological effects of long-term and overuse of manure on farmland soils. Our aim is to draw attention to the importance of research on the combination of breeding and breeding. Indeed, the description of lines 43-44 may be misleading to the reader, so, after careful consideration, we have removed it from the manuscript. You can see our changes on line 389-390.

Comment 9. Line 53 - please revise the term " people's fertilization habits". This should refer to the recommendations being provided based on research.

Response: Thank you for your suggestions. After careful consideration, we have dropped the term " people's fertilization habits". You can see our changes on line 396.

Comment 10. Line 58 and throughout the manuscript- please correct the term "researches". More research or more studies should be conducted.

Response: The term "researches" has been changed to "Many scholars have studied…".  And the other "researches" has been changed to “More research should be conducted”. You can see our changes on lines 49-50, 67.

Comment 11. Line 61- please remove the ";" and start a new sentence after it.

Response: Kindly yours, we have already removed the ";". You can see our changes on line 70.

Comment 12. Line 86- "was" not "were"

Response: Kindly yours, we have changed "was" to "were". You can see our changes on lines 85, 87.

Material and methods

Comment 13. Did the authors developed these parameters, or are they using parameters that are well established? If the latter, please use references to previous studies using those parameters. If they are new parameters, please use references showing how those were thought about to be developed and to back up the line of thought.

Response: Thank you for your sincere suggestions, we mainly used the methods involved in previous studies. After summarizing the previous methods, we got the strategic formula for nutrient allocation. All the parameters in the formula are derived from the calculation results of previous research formulas. We have checked all the formula parts and have covered all the relevant references and formula interpretation.

Comment 14. Line 117 - why are the authors basing this parameter in "pig equivalent". Also, please make sure to determine what emissions the authors are referring to.

Response: To compare regional differences in emissions of different livestock and poultry manure at the same standard, we used pig equivalent as a conversion indicator. We have already added the explanation and the definition of emissions in the text. The emission has been changed to “the emission of livestock and poultry manure in terms of pig manure equivalent”. You can see our changes on lines 114-115, 118, 124-125.

Comment 15. Line 171 - please include a reference

Response: The reference has been added. You can see our changes on line 172.

Comment 16. Lines 198-200 - Based on this statement, the authors did not conduct any statistical analysis of the response values obtained. Is that correct?

Response: Thank you for your valuable question. We have conducted the statistical analysis using these formulas in the text. But we did not evaluate the response values obtained from the formula. This can be explained by several reasons. First of all, our analytical formulas are basically derived from previous studies, which have been proven to be usable. Secondly, data analysis is comparative analysis under the same standard, aiming to find out the area with high fecal emission and analyze its overload risk. Finally, for the screened overloading area, we can provide a theoretical overloading and nutrient allocation. Because we aim to provide a method to calculate the overloading and nutrient allocation. Thus, we can provide a method to calculate the theoretical value and distribution path of nutrient allocation. We have already added the “All statistical analysis data were calculated using the above formulas” in the text. You can see our changes on line 204-205.

Comment 17. All figures need to be revised to provide better visual aid and the legend of some need to be revised as well. Major revisions are required throughout the manuscript to improve the results and the discussion sections.

Response: Thank you for your sincere suggestions. We have provided visible-legend figures in the text. We guarantee that all the figures in the manuscript are good-visible. Meanwhile, we keep the legend and direction arrow of each figure, because we think it is a kind of symmetry beauty. If one or two direction arrows are deleted, they always show disharmony. You can see our changes on lines 268-269, 276-277, 287-288. And we have improved the results and the discussion sections. The literature citations mainly included in the result part have been put into the introduction and discussion, and other unavailable information has been deleted. You can see our changes on lines 46-48, 66-67, 325-326, 412-414.

Reviewer 2 Report

The authors made significant changes to the manuscript and addressed all my suggestions during the review of their manuscript. I think that the manuscript could be considered for publication after some very minor changes:

Figure 2: I recommend to define the (a), (b) and (c) points in alphabetical order as it is confusing this present notation (a, c, b)

Conclusion: I recommend to further synthesize this section because it includes to many details and, as such, the main findings of the manuscript are somehow lost between lines. I would give up the first paragraph or synthesize it.

Author Response

General response: We sincerely thank the editor and all reviewers for their valuable feedback that we have used to improve the quality of our manuscript. The reviewer comments are laid out below and specific concerns have been numbered. Our response is given in normal font and changes/additions to the manuscript are given in red text.

Response to Reviewer

Comment 1. Figure 2: I recommend to define the (a), (b) and (c) points in alphabetical order as it is confusing this present notation (a, c, b)

Response: Thank you for your sincere suggestions. We have modified the alphabetical order in Figure 2 into the (a), (b) and (c) points and also changed it in the text. You can see it on line 213-218.

Comment 2. Conclusion: I recommend to further synthesize this section because it includes to many details and, as such, the main findings of the manuscript are somehow lost between lines. I would give up the first paragraph or synthesize it.

Response: Thanks for your sincere suggestions. These suggestions are very important for the revision of this paper and the guidance of follow-up research. According to your suggestion, we think it is a wise choice to delete the first paragraph of the conclusion. So, we removed the first paragraph from the conclusion and synthesized the conclusions in the form of every result for every small conclusion, and the last big conclusion is a summary. For example, the first point was changed to “Livestock manure emissions were higher in the southeastern and northwestern parts of the study area than in the middle. However, the nitrogen requirement of crops showed a decreasing trend from north to south, suggesting that risk of contamination may exist in the south because of the livestock and poultry waste discharge.” You can see our changes on line 430-445.
